# Three-dimensional mapping reveals heterochronic development of the neuromuscular system in postnatal mouse skeletal muscles

Jianyi Xu[1,2,3], Jingtan Zhu[1,2,3], Yusha Li[1,2], Yingtao Yao[1,2], Ang Xuan[1,2], Dongyu Li[1,2], Tingting Yu ⬤ [1,2✉] & Dan Zhu ⬤ [1,2✉]

The development of the neuromuscular system, including muscle growth and intramuscular neural development, in addition to central nervous system maturation, determines motor ability improvement. Motor development occurs asynchronously from cephalic to caudal. However, whether the structural development of different muscles is heterochronic is unclear. Here, based on the characteristics of motor behavior in postnatal mice, we examined the 3D structural features of the neuromuscular system in different muscles by combining tissue clearing with optical imaging techniques. Quantitative analyses of the structural data and related mRNA expression revealed that there was continued myofiber hyperplasia of the forelimb and hindlimb muscles until around postnatal day 3 (P3) and P6, respectively, as well as continued axonal arborization and neuromuscular junction formation until around P3 and P9, respectively; feature alterations of the cervical muscle ended at birth. Such structural heterochrony of muscles in different body parts corresponds to their motor function. Structural data on the neuromuscular system of neonatal muscles provide a 3D perspective in the understanding of the structural status during motor development.

[1] Britton Chance Center for Biomedical Photonics - MoE Key Laboratory for Biomedical Photonics, Wuhan National Laboratory for Optoelectronics - Advanced Biomedical Imaging Facility, Huazhong University of Science and Technology, 430074 Wuhan, Hubei, China. [2] Optics Valley Laboratory, 430074 Wuhan, Hubei, China. [3]These authors contributed equally: Jianyi Xu, Jingtan Zhu. ✉email: yutingting@hust.edu.cn; dawnzh@mail.hust.edu.cn

Motor performance is one of the most notable features after birth, providing functional information for the determination of neuromuscular development, with structural integrity dictating normal function. Every progress made during motor development depends on the fundamental structural characteristics of the neuromuscular system[1,2]. Motor development is primarily driven by the maturation of the central nervous system, but dramatic changes in infants' hardware, such as muscle size and strength, also play an essential role[3]. Comprehensively understanding muscle growth and intramuscular neural development should undoubtedly contribute to the understanding of neuromuscular development.

Motor development is asynchronous and proceeds from cephalic to caudal according to the developmental principle[3,4]. For example, infants gradually become able to lift their heads, crawl, stand, and walk, and arm movement always comes before leg movement during gross motor development. However, it is unclear whether structural development is heterochronic in developing muscles from different body parts.

Most inquiries made on the structural features of the neuromuscular system have used experimental animal models[5-7], with mice among the most widely used animals to model human infants because of their immaturity at birth and slow ontogenic processes of the nervous system during postnatal development[8,9]. Skeletal muscles – locomotive organs – consist of contractile myofibers. An increase in myofiber numbers (hyperplasia) and/or expansion of individual myofibers (hypertrophy) can spur muscle growth. Postnatal skeletal muscle growth is rapid, with total muscle mass increasing significantly in the first few weeks, providing enough support for motor function development. However, there is no clear consensus among scholars on whether there is myofiber hyperplasia in skeletal muscles, i.e., an increase in myofiber number, during postnatal development. Some studies have suggested that total myofiber numbers are determined during embryogenesis, but others have demonstrated that fiber numbers in some muscles continue to increase after birth[10-14]. This controversy over myofiber hyperplasia confuses the understanding of structural development in postnatal muscles.

In addition to muscle growth, the development of the neuromuscular junction (NMJ), a chemical synapse between neurons and myofibers, is essential to improving muscle function[15,16]. NMJs are formed initially during embryonic growth and undergo a series of maturation processes, including morphological complication and synapse elimination, during early postnatal development[17-19]. These processes refine NMJs' structural organization and enhance neurotransmission efficiency to facilitate motor performance during postnatal development. Most investigations have focused on developing NMJs' morphological changes in the longitudinal muscle sections, isolated myofibers, or flat muscles using traditional histological methods, and this strategy has played a crucial role in understanding NMJ maturation[20-22]. However, for muscle types that are widely distributed and numerous in the entire body, limited muscle selection can easily result in a biased appraisal of the development of NMJs. Moreover, NMJs are distributed unevenly in intact skeletal muscles, and their 3D spatial distributions are closely associated with motor functioning[23,24]. The lack of information on the spatial distribution of NMJs in developing muscles makes it difficult to associate NMJ growth with functional improvement. In a nutshell, based on the existing structural information on developing muscles, the developmental heterochrony in different muscles has not been demonstrated, thereby limiting the understanding of the relationship between structure and function.

In recent years, combining tissue optical clearing with optical imaging techniques has allowed whole-mount analyses of the morphology, number, and distribution of NMJs in whole muscles, including large-volume muscles, such as the gastrocnemius and quadriceps, providing a new approach to study the development of the neuromuscular system[25-27].

Here, we scrutinized the structural development of the neuromuscular system in different muscles from a 3D perspective, established its heterochrony in different body parts, and determined the correlation between structural development and motor development. Motor performances in different body parts of postnatal mice were characterized using behavioral testing, and 3D structural features of the neuromuscular system in cervical, forelimb, and hindlimb muscles during postnatal development were obtained via the combined use of the FDISCO method and light-sheet microscopy. Changes in myofiber, nerve branch, and NMJ growth at the overall level of the muscle were quantitatively analyzed. The development of axon terminals was evaluated by detecting collagen XXV mRNA expression using qRT-PCR.

## Results

**Heterochronic motor development in neonatal mice**. Neonatal mice develop rapidly during the first 2 weeks of life after birth, so we selected P0, P3, P6, and P9 mice in this study. The weights of mice increase from 1.25 g on P0 to 4.53 g on P9 (Fig. 1a). During the periods selected above, we evaluated the righting response of pups and used the time taken by the pups to turn over as an indicator to assess their overall motor abilities, including strength and coordination. Almost all P0 and P3 pups failed to right within 20 s, although they attempted to turn over by rolling and struggling (Fig. 1b). P6 pups righted themselves in 6.7 s, while P9 pups, undoubtedly aided by their motor improvement, righted themselves in less than 1 s, just like adult mice.

Limb function was assessed using the limb placement response technique (Fig. 1c). Score distributions for the forelimb and hindlimb placing responses are displayed in the violin plot (Fig. 1d). The occurrence of the forelimb placing response began around P3, whereas that of the hindlimb placing response started around P6 and stabilized on P9. These findings indicate that the functional development of the hindlimb lags behind that of the forelimb.

We used the walking pattern (Fig. 1e, f) and cliff avoidance (Fig. 1g, h) to measure pups' motor performance in the open-field and dangerous conditions, respectively, further determining the developmental progress of limb motor function at each time point. Almost all P0 pups were static in the open-field test, but they often raised their heads. The pups' reaction at the edge of a cliff was to turn their heads away to avoid the cliff. The forelimbs of the P3 pups produced a paddling motion that allowed them to pivot in the open field and turn away from the cliff. In addition to stable pivoting motions, P6 pups began to crawl with a stagger in the open field using their limbs, including the hindlimbs; however, upon facing the cliff, they relied heavily on the strength of their forelimbs to avoid it. On P9, the development of the hindlimb function provided the main propulsive force, aiding the pups to walk like adults and retract their bodies to avoid the cliff upon reaching its edge.

Per the above results, the motor development of mice during postnatal development is sequential and heterochronic, i.e., the motor development of the cervical parts occurs earlier than that of forelimbs, and the motor development of forelimbs takes place before that of hindlimbs. Cervical muscles can support head movements at birth, but the limbs, still weak at birth, cannot move the body. The forelimbs become functional on P3, producing the circle movement, while the hindlimbs begin to participate in movement on P6 but are still weak and cannot support the body and maintain balance. By P9, the hindlimbs function have developed to provide propulsive force for stable quadruped walking.

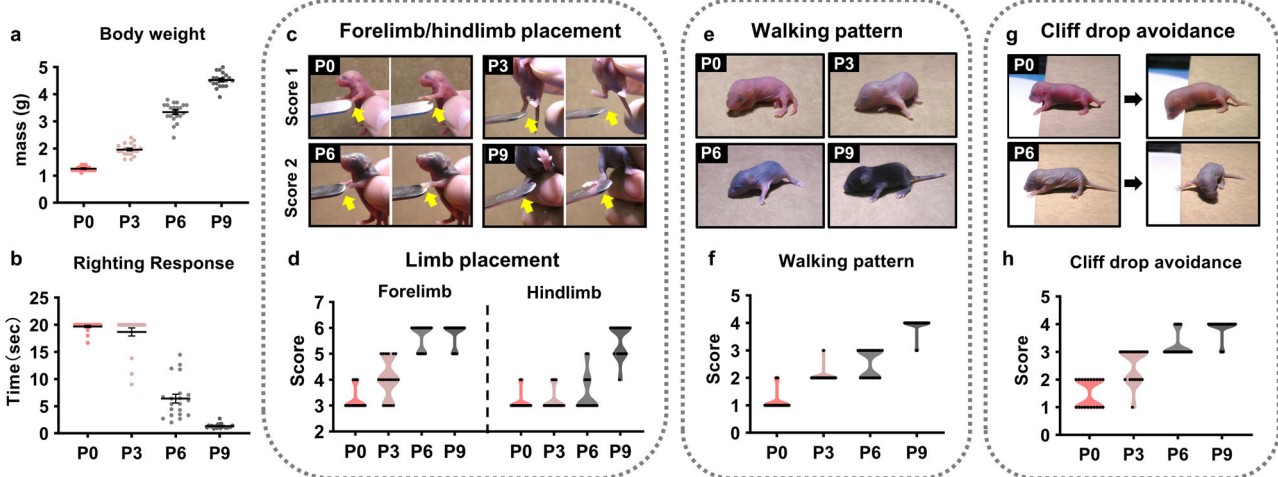

**Fig. 1 The measurement of motor development in neonatal mice. a** Weight (in grams) of postnatal mouse pups. **b** The average time taken to complete the righting response. When a pup failed to reach the adult pattern within 20 s, its time was recorded as 20 s. **c** The test for the limb placing response. **d** The scores for forelimb and hindlimb placing responses after birth. **e** The walking pattern in the open-field experiment during postnatal development. **f** The scores for the characteristics of the walking pattern at different time points. **g** The test for cliff drop avoidance. **h** The scores for the motor performance test when the pups faced the cliff at different time points. $n = 20$ independent animals for each behavioral test.

**Postnatal skeletal muscle fiber development.** We selected common experimental muscles in from different body parts, including the cleidomastoid, biceps brachii, gastrocnemius, and tibialis anterior for this study (Fig. 2a and Supplementary Fig. 1a). The comparisons of the muscle sizes at different times after birth qualitatively demonstrated the rapid progression of early postnatal development.

Muscle fibers are formed when myoblasts fuse to produce multinucleated myotubes that can further mature into myofibers[28,29]. The developmental stage of muscle fibers can be determined preliminarily based on the position of myonuclei in myofibers. In this study, we used anti-Dystrophin antibody and nuclear staining (DAPI) to label myofiber membranes and myonuclei in the muscle sections. The co-labeled images in Fig. 2b show the relative positions of myofiber membranes and myonuclei in different muscles during postnatal development and adulthood. According to findings from the quantification of the percentage of myofibers with central nuclei (Fig. 2c), central myonuclei rarely occupied the cleidomastoid areas after birth (4.60%), with most migrating to the periphery, pointing to the completion of myogenesis. As for the biceps brachii and gastrocnemius on P0, the percentages of myofibers with central myonuclei were 19.57% and 34.49%, respectively, indicating that some fibers in these two muscles were still immature myotubes. Subsequently, the central nuclei almost disappeared in the biceps brachii on P3 (3.93%) and in the gastrocnemius on P6 (2.38%), showing that myotubes in the biceps brachii mature before those in the gastrocnemius.

During myogenesis, myoblasts are highly proliferative prior to fusion and exit the cell cycle after fusion[30]. Therefore, quantifying proliferative cells in developing muscles should provide a measurable parameter for the determination of muscle development. Here, we used the anti-Mki67 antibody to stain the proliferative cells in the muscle sections and analyzed the percentages of the proliferative cells (KI67 positive cells, KI67+) quantitatively in different muscles during postnatal development and adulthood (Fig. 2d, e). No KI67 + cells were identified in adult skeletal muscles, an expected occurrence because satellite cells, as muscle stem cells, are silent in normal adult muscles. However, during postnatal development, especially within the first three days of birth, the percentages of proliferating

cells (KI67+) in different muscles varied significantly. The percentages of proliferating cells in the gastrocnemius (49.99%) and biceps brachii (37.14%) on P0 were considerably higher than that in the cleidomastoid (13.95%) but decreased to low levels on P6 (15.85%) and P3 (15.33%). These quantitative results provide further evidence of developmental heterochrony in skeletal muscles from different parts of the body. Notably, myofibers in the gastrocnemius on P0 were immature, with a higher percentage of proliferating cells (50%) than on P6 and P9 (~15%), suggesting that myofiber formation potentially still occurs after birth, not just myofiber hypertrophy.

We quantified myofiber numbers in developing and adult skeletal muscles. To ensure accurate quantification, 100 μm thick transverse sections of the muscle belly were used. The clear outlines of myofiber membranes stained with the anti-Dystrophin antibody allowed us to quantify the fibers accurately (Fig. 2f and Supplementary Fig. 1b). We found no significant changes in myofiber numbers in the cleidomastoid from birth to adulthood (Fig. 2g). Fiber numbers in the biceps brachii (Fig. 2h) increased from $2541 \pm 83$ on P0 to $3077 \pm 67$ on P3, suggesting that myofiber formation goes on after birth but ends before P3. Similarly, myofiber formation in the gastrocnemius (Fig. 2i) and tibialis anterior (Supplementary Fig. 1c) occurs after birth but ends before P6.

All these findings suggest that the development of skeletal muscles in different body parts is asynchronous. In conjunction with motor function development, limb motor function improvement depends on the hypertrophy of muscle fibers and an increase in myofiber numbers during the early postnatal period.

**3D visualization of innervating nerves and NMJs in developing skeletal muscles.** The entire process of visualizing innervating nerves and NMJs is shown in Fig. 3a. The in vivo injection of *Thy1*-YFP-16 mice with fluorescent α-BTX enables intramuscular motor nerves and NMJs to be whole-mount labeled. The muscle samples were cleared using the FDISCO method, imaged utilizing light-sheet microscopy, and reconstructed with Imaris to analyze the 3D distribution of intramuscular nerve branches and NMJs during postnatal development.

Each skeletal muscle had a unique innervation pattern, one that remains roughly identical from birth to adulthood (Fig. 3b–d and

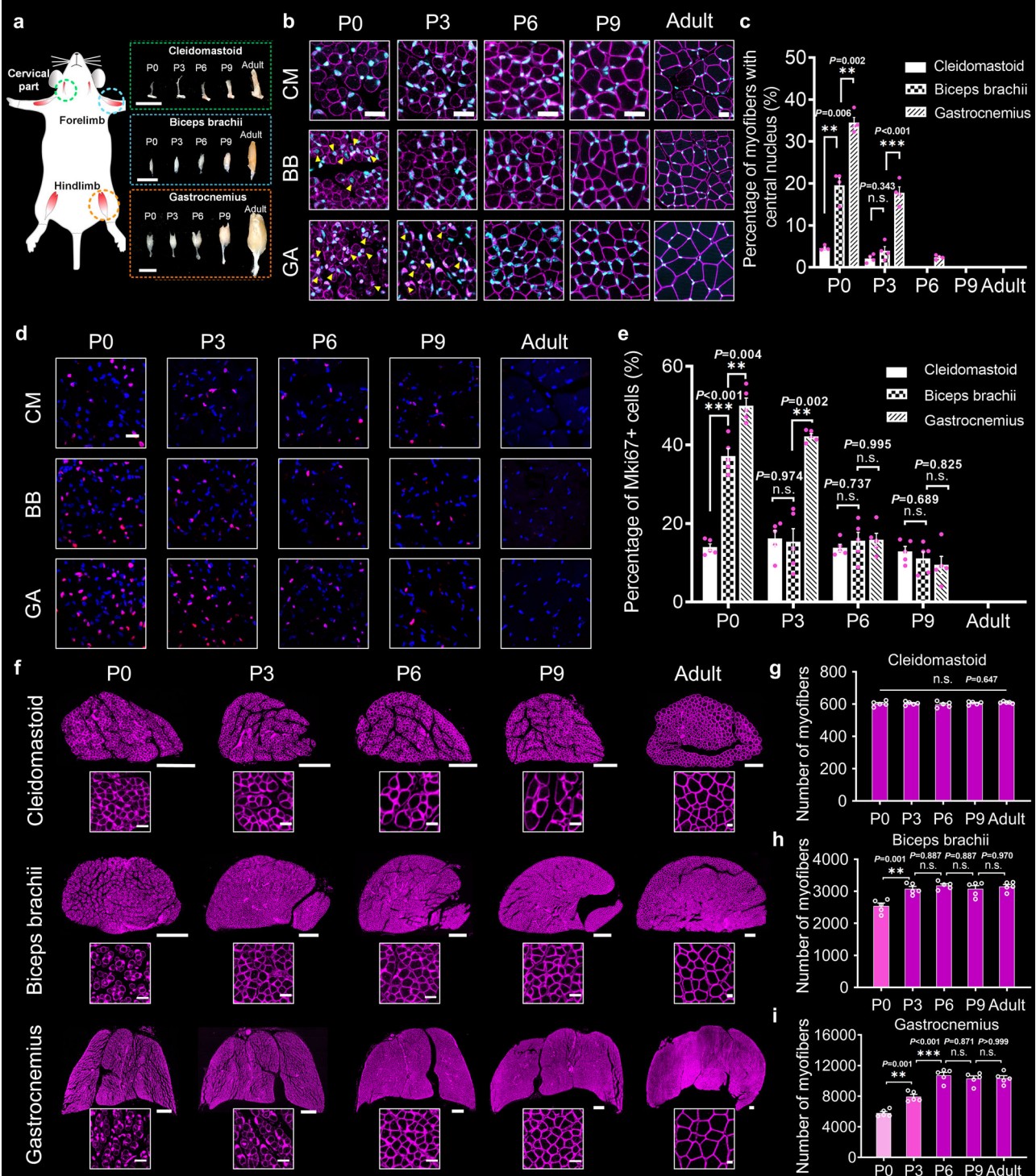

**Fig. 2 Postnatal development of myofibers in different skeletal muscles. a** The selection of muscles from different body parts, including the cleidomastoid (CM) in the cervical part, the biceps brachii (BB) in the forelimb, and the gastrocnemius (GA) in the hindlimb. Scale bar: 5 mm. **b** Fluorescence imaging of the location of myonuclei (cyan) relative to the membrane (magenta) in the cross-sections of different muscles after birth. The yellow arrowheads indicate myonuclei located at the center of myofibers. Scale bar: 25 μm. **c** The quantification of the percentage of myofibers with central nuclei in **b** (*n* = 4 independent animals for each muscle at each time point). **d** Fluorescence imaging of proliferative cells (magenta) in the cross-sections. Scale bar: 25 μm. **e** The quantification of the percentage of KI67 + cells in **d** (*n* = 5 independent animals for each muscle at each time point). **f** Fluorescence imaging of myofiber membranes in the cross-sections; scale bar: 200 μm. The images in the white frames show the detailed morphologies of myofibers; scale bar: 25 μm. **g–i** The quantification of myofiber numbers in the cleidomastoid (**g**), biceps brachii (**h**), and gastrocnemius (**i**) (*n* = 5 independent animals for each muscle at each time point). All values are presented as the means ± SEM; statistical significance in **c** and **e** (n.s. represents not significant, **\**P* < 0.01, and \*\*\**P* < 0.001) was assessed using two-way ANOVA followed by Tukey post hoc test; statistical significance in **g–i** (\*\**P* < 0.01 and \*\*\**P* < 0.001) was assessed using one-way ANOVA followed by Tukey post hoc test.

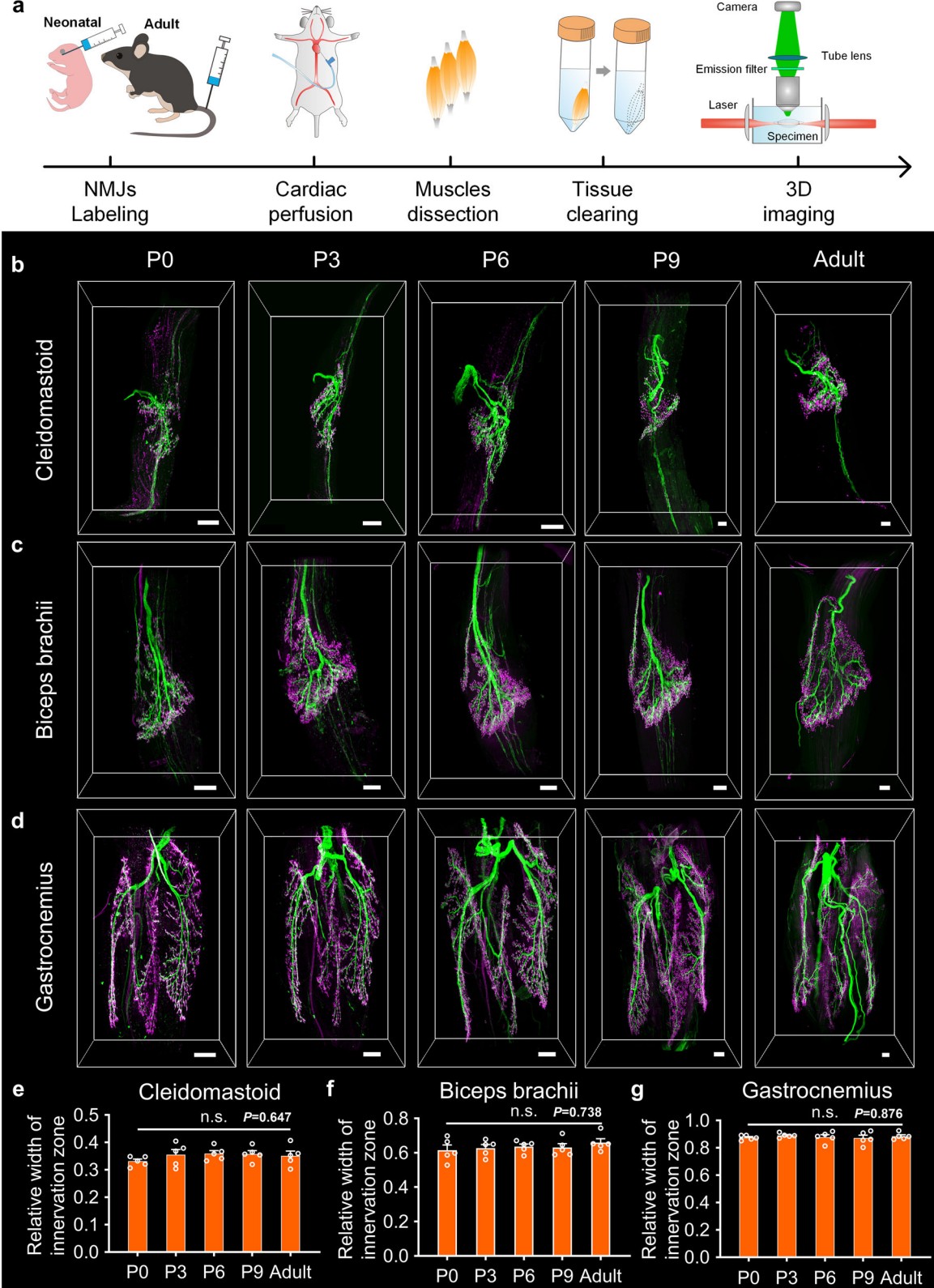

**Fig. 3 3D visualization of innervating nerves and NMJs in skeletal muscles after clearing. a** The entire experimental process of the 3D imaging of intramuscular motor nerves and NMJs in skeletal muscles. **b–d** 3D reconstructions of intramuscular motor nerves and NMJs in the cleidomastoid (**b**), biceps brachii (**c**), and gastrocnemius (**d**) during postnatal development and adulthood. Scale bar: 200 μm. **e–g** The quantification of the relative ranges of innervation zones in whole muscles ($n = 5$ independent animals for each muscle at each time point). All values are presented as the means ± SEM; statistical significance in **e–g** (n.s. represents not significant) was assessed using one-way ANOVA.

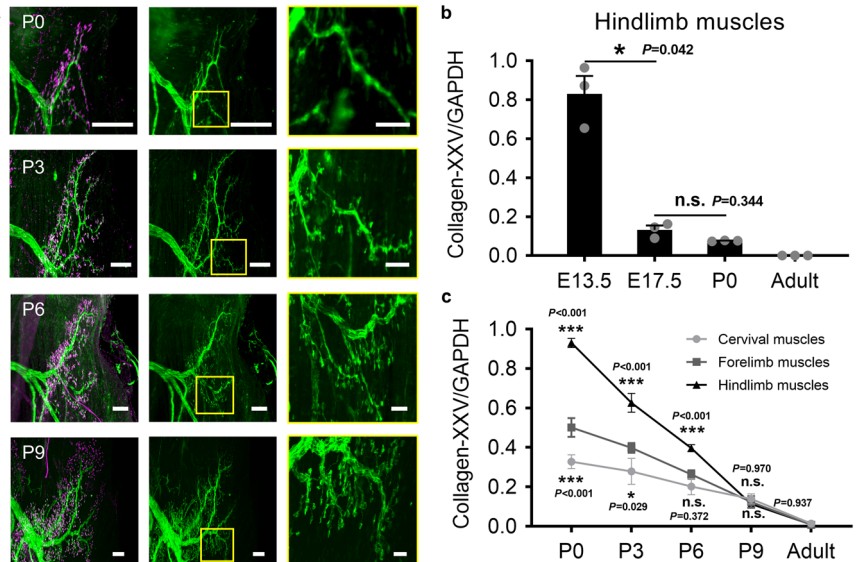

**Fig. 4 Intramuscular axonal arborization during postnatal development. a** A comparison of the motor nerve arborizations in the gastrocnemius during postnatal development. The images in the yellow frames in the second column are enlarged and presented in the third column: they show increased axonal terminals and branches on the selected branch during postnatal development. Scale bar in white frames: 100 μm; scale bar in yellow frames: 30 μm. **b** qRT-PCR analysis of collagen XXV mRNA expression in embryonic (E13.5 and E17.5), neonatal (P0), and adult hindlimb muscles ($n = 3$ independent animals for each time point). mRNA levels were normalized to GADPH expression. **c** qRT-PCR analysis of collagen XXV mRNA expression in postnatal and adult muscles from the cervical part, forelimb, and hindlimb ($n = 6$ independent animals for each body part at each time point). All values are presented as the means ± SEM; statistical significance in **b** and **c** (n.s. represents not significant, *$P < 0.05$, and ***$P < 0.001$) was assessed using one-way ANOVA and two-way ANOVA, and followed by Dunnett's T3 post hoc test and Tukey post hoc test respectively.

Supplementary Fig. 2a). The cleidomastoid had a simple distribution pattern that concentrates around the mid-point or in the superior portion (Fig. 3b). The distribution pattern in the biceps brachii was lamellar parallel to the muscle (Fig. 3c) and was divided into two regions that innervate the long head and the short head, respectively. The tibialis anterior is larger and had a more complex innervation pattern, like a triangular pyramid (Supplementary Fig. 2a). The gastrocnemius, with more than ten thousand myofibers, displayed a complex innervation pattern, with its innervation zone distributed in the whole muscle (Fig. 3d).

We quantitatively compared the distribution ranges of the innervation zones during postnatal development by analyzing the relative width of the entire muscle (Supplementary Fig. 2b). The innervation zone of the cleidomastoid was concentrated, and its width accounted for only 35.13% of the entire width of the adult muscle. There was no significant difference in the distribution range of the innervation zone in the cleidomastoid relative to the entire muscle from birth to adulthood (Fig. 3e). In the biceps brachii (Fig. 3f) and tibialis anterior (Supplementary Fig. 2c), the relative ranges of the innervation zones were both about 60% and remained constant after birth. The relative width of the innervation zone of the gastrocnemius remained constant at about 90% from birth to adulthood, covering almost the entire muscle (Fig. 3g). All these results indicate that the enlargement of the innervation zone is in synchrony with the growth of muscles to ensure improvement in muscle function during postnatal development.

**Intramuscular axonal arborization in postnatal muscles.** Motor nerve arborizations in the upper region of the medial gastrocnemius head during postnatal development were compared (Fig. 4a). Nerve arborization and terminals on the branch in the yellow box were both sparse on P0. Further development (shown in the images on P3) triggered the appearance of several new axon

terminals on the branches. On P6, the branches were elongated and enlarged, and more branches appeared, enhancing the complexity of nerve arborization. On P9, nerve arborization was abundant, and axon terminals were dense. These findings suggest that axonal arborization in the gastrocnemius continues to occur within the first few days of birth, providing new axon terminals and branches for intramuscular motor innervation.

Collagen XXV is transmembrane-type collagen that regulates axonal arborization and is indispensable for motor innervation in developing muscles[31]. Its mRNA is expressed in developing skeletal muscles and decreases gradually with the maturation of muscles, reflecting the progress of the axonal arborization of intramuscular motor nerves. We first assessed the level of collagen XXV mRNA expression in hindlimb muscles at different developmental stages. The results in Fig. 4b confirmed that collagen XXV mRNA expression was abundant in embryonic hindlimbs and decreased gradually with continued development. The result also demonstrated that postnatal hindlimb muscles retained a certain level of collagen XXV mRNA expression, which regulates axonal arborization in hindlimb muscles after birth.

Collagen XXV mRNA expression was compared further between muscles from the cervical part, forelimbs, and hindlimbs during postnatal development. Figure 4c revealed that collagen XXV mRNA expression differed significantly between these muscles at birth; however, these differences were not notable on P9. These findings demonstrate the heterochronic development of intramuscular motor nerves in different body parts early after birth. Collagen XXV mRNA was least expressed in cervical muscles at birth, indicating that intramuscular nerves in cervical muscles were most developed compared to those in the forelimb and hindlimb muscles. And during subsequent development, the only slight decrease in mRNA expression in addition to the myofiber development of the cervical cleidomastoid suggested that axonal arborization in the cervical muscles is probably almost completed at birth. Using the expression in cervical

muscles at birth as a reference, we further deduced that axonal arborization in the forelimb muscles continued after birth and was completed most likely between P3 and P6, whereas axonal arborization in the hindlimb muscles ended between P6 and P9. In a nutshell, intramuscular motor nerve development among muscles in different body parts is heterochronic, with the occurrence of axonal arborization in forelimb and hindlimb muscles continuing after birth to achieve intramuscular motor innervation.

**The consistency of intramuscular innervation patterns after birth**. Axonal arborization provides new terminals and branches for motor nerve innervation. However, whether axonal arborization affects the intramuscular innervation pattern during postnatal development was unknown. Therefore, we analyzed the innervation pattern in whole-mount muscles based on 3D reconstruction.

The raw fluorescent images of the nerve branches of the cleidomastoid (P0), biceps brachii (P6), and gastrocnemius (P3) are shown in Fig. 5a–c. Conducting 3D tracing using the Imaris software, we located nerve branches of different sizes; the primary branches are marked with different colors (Fig. 5d–f and Supplementary Movies 1–3). In the cleidomastoid, the nerve entered the muscle and separated into three primary branches, each of which independently innervated one region (Fig. 5d). The biceps brachii was also innervated by three primary nerves extending from the proximal to the distal; two of the branches were larger and controlled the long head, while the third branch was smaller and was responsible for the short head (Fig. 5e). The innervation pattern of the gastrocnemius was divided into three portions consisting of twelve primary branches (Fig. 5f).

Further analyses of the spatial distribution and the number of primary branches in each muscle during development revealed that the distribution patterns of the primary branches were almost identical from birth to adulthood (Supplementary Figs. 3–5 and Fig. 5j–l). Subsequently, we selected a single primary branch from each muscle for additional evaluation of secondary branches during development (Fig. 5g–i). Our quantitative results showed no significant difference in the number of secondary branches at different developmental times (Fig. 5j–l), with the distributions of secondary branches also substantially the same (Supplementary Figs. 3–5). These findings suggest that the number of primary and secondary branches in the cleidomastoid, biceps brachii, and gastrocnemius remain unchanged after birth, and the innervation pattern in each muscle remains constant during postnatal development.

**NMJ formation in skeletal muscles during postnatal development**. Based on the results in Fig. 4, NMJ formation is potentially still ongoing in some muscles during postnatal development. Therefore, we evaluated NMJ development via a whole-mount analysis of skeletal muscles.

Using the gastrocnemius as an example, the 3D distributions of NMJs in the developing and adult muscles are shown in Fig. 6a. The fluorescent signals are the AChR (acetylcholine receptor) clusters labeled with the α-BTX, each of which was used to represent an individual NMJ. The enlarged images of the corresponding positions in the white frames and the segmentation result show the differences in the density of fluorescent signals throughout postnatal development. NMJ numbers in the gastrocnemius were $4296 \pm 304$ on P0, increasing to $9913 \pm 374$ on P6, $11676 \pm 149$ on P9, and $12089 \pm 175$ in adults, indicating that NMJ formation goes on in the gastrocnemius after birth, possibly reaching completion between P6 and P9 (Fig. 6b).

Analyses of NMJ numbers in the cleidomastoid, biceps brachii, and tibialis anterior during postnatal development (Supplementary Fig. 6a–c) revealed similar changes in NMJ numbers in the tibialis anterior located in the hindlimb to those of the gastrocnemius (Supplementary Fig. 6d). NMJ numbers in the biceps brachii on P0 were $1576 \pm 137$, increasing to adult levels on P3 (Fig. 6c), indicating that NMJ formation completion in the biceps brachii possibly occurs before P3. NMJ numbers in the cleidomastoid did not differ significantly between birth and adulthood (Fig. 6d), suggesting that NMJ formation in the cleidomastoid is completed before birth.

Based on these findings, the difference in the time during which NMJ numbers reach adult levels reflects the heterochrony of NMJ development in skeletal muscles of different parts of the body. Moreover, alongside motor performance after birth, NMJ formation in limb muscles suggests that motor improvement during early postnatal development depends on the maturation of NMJ morphology and numbers in whole muscles.

**Discussion**
In this study, we examined the structural development of the neuromuscular system in the skeletal muscles of mice, including myofiber hyperplasia, axonal arborization, and NMJ formation, at the early postnatal stage. We identified the heterochronic development of the neuromuscular system corresponding to motor development.

Specific skeletal muscles were selected in this study according to their distribution in the anteroposterior (A-P) axis, including the cleidomastoid, biceps brachii, gastrocnemius, and tibialis anterior, which are all 'fast' muscles dominated by type II fibers[32–34]. 'Slow' muscle types were not examined in this investigation. In addition, other important parameters, such as the distance between the muscle and the spinal cord (proximodistal axis)[35], contribute to developmental characteristics and deserve to be scrutinized in future work.

Analysis of muscle development helps understand motor development during postnatal growth. However, past conclusions on muscle development after birth are inconsistent. Two factors may explain these inconsistent results: one is the difference in the respective muscles assessed between studies due to the heterochronic development of skeletal muscles in various body parts. The other factor is the selection of time points in various investigations. Ross et al. found that myofiber numbers in the IVth lumbrical muscle in the hindlimbs increase to adult levels on P7[13], and we also showed that the increase in myofiber numbers is completed within the first week of birth. On the other hand, Ontell et al. and White et al. both reported no changes in total myofiber numbers during the postnatal development of the extensor *digitorum longus* in the hindlimbs; however, these findings were based on P7 and P14 data, both periods after the increase in myofibers has come to completion[11,36]. Therefore, both the selection of muscle types and time points is essential to exploring muscle development and making proper inferences.

During myofiber hyperplasia, proliferative myoblasts exit the cell cycle after fusion[30]. Therefore, the number change in proliferative cells was used to assess the developmental level of different muscles in this study. Alongside the myofiber number results, we found the existence of about 10–20% of proliferative cells in three skeletal muscles when myofiber numbers developed to adult levels. Considering that myofiber hypertrophy requires an increase in myonuclei number within the first 21 days of birth[11], it can be speculated that these proliferative cells are necessary to provide new myonuclei for supporting myofiber hypertrophy. A percentage of proliferative cell numbers significantly higher than 20%, such as in the gastrocnemius on P0

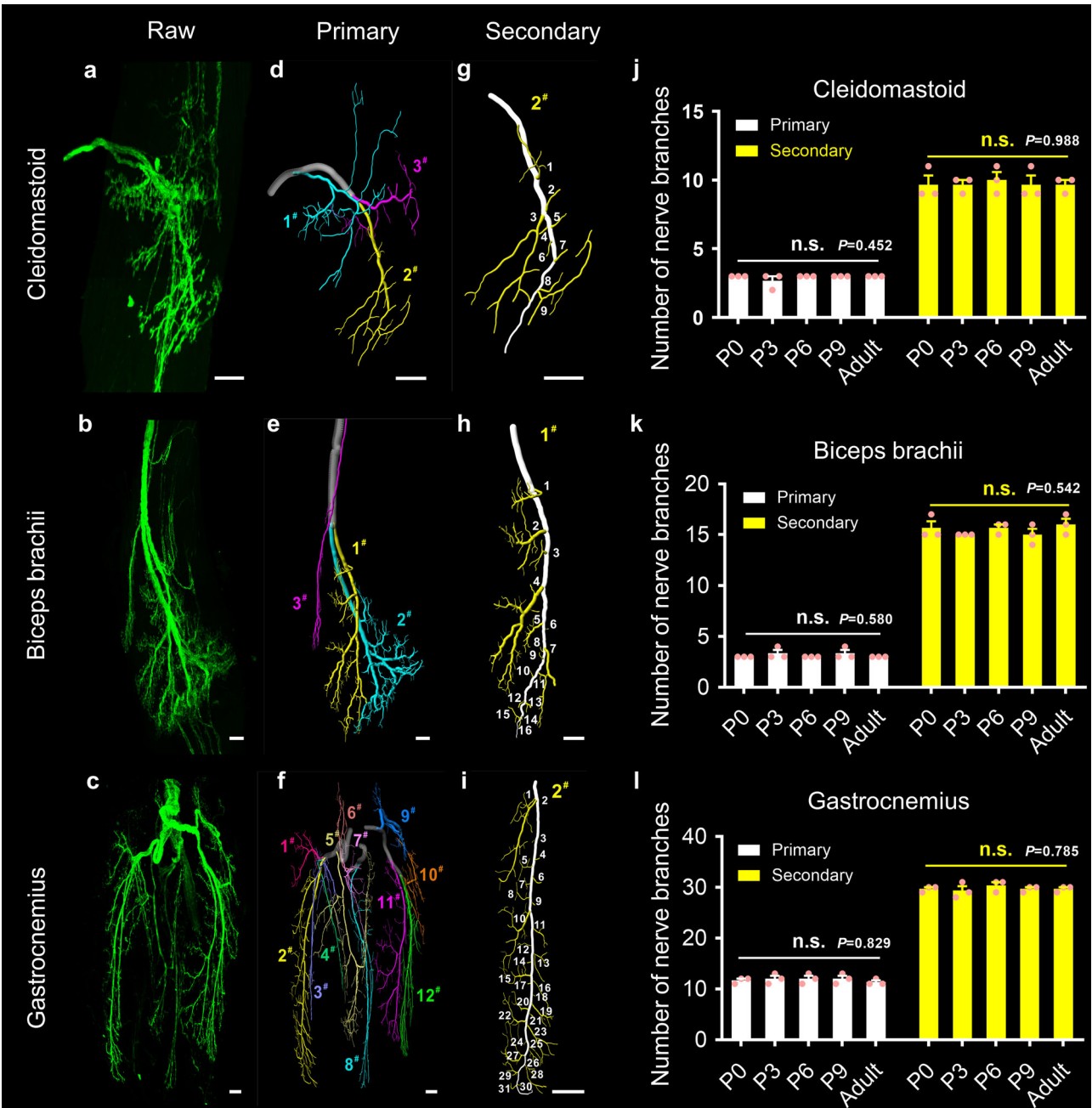

**Fig. 5 3D analysis of the intramuscular innervation pattern during postnatal development. a–c** The spatial conformation of intramuscular motor nerves in a P0 cleidomastoid (**a**), P6 biceps brachii (**b**), and P3 gastrocnemius (**c**). **d–f** The segmentation of the nerve branches in **a–c** by 3D tracing using the Imaris software. The primary branches were noted as originating from the nerve branches entering the muscles and extending from proximal to distal, and they were numbered and marked with different colors. **g–i** The distribution of secondary branches in the 2# primary branch of the cleidomastoid (**g**), 1# primary branch of the biceps brachii (**h**), and 2# primary branch of the gastrocnemius (**i**). **j–l** The quantification of the primary and secondary branches in the cleidomastoid (**j**), biceps brachii (**k**), and gastrocnemius (**l**) during postnatal development and adulthood ($n = 3$ independent animals for each muscle at each time point). Scale bar: 100 μm. All values are presented as the means ± SEM; statistical significance in **j–l** (n.s. represents not significant) was assessed using one-way ANOVA.

and P3, points to the occurrence of not only myofiber hypertrophy but also myofiber hyperplasia. Thus, the quantitative analysis of proliferative cells not only compares the differences in developmental levels between different skeletal muscles but also provides a reference for determining muscle hypertrophy and hyperplasia.

The emergence of tissue optical clearing has provided a powerful tool for 3D imaging of biological tissues[37,38]. In recent years, many clearing methods have been developed, including MACS[39], PEGASOS[40], and FDISCO[41]. Each technique has its advantages in coping with variable experimental conditions. FDISCO allows the 3D visualization of nerve branches and NMJs in adult skeletal muscles, and its dehydration step improves the hardness of samples, contributing to the easier mounting of the clearing muscles on the sample holder. Thus, we used the FDISCO method in this study and further modified the clearing time according to the sizes of the neonatal muscles to preserve the fluorescence as much as possible.

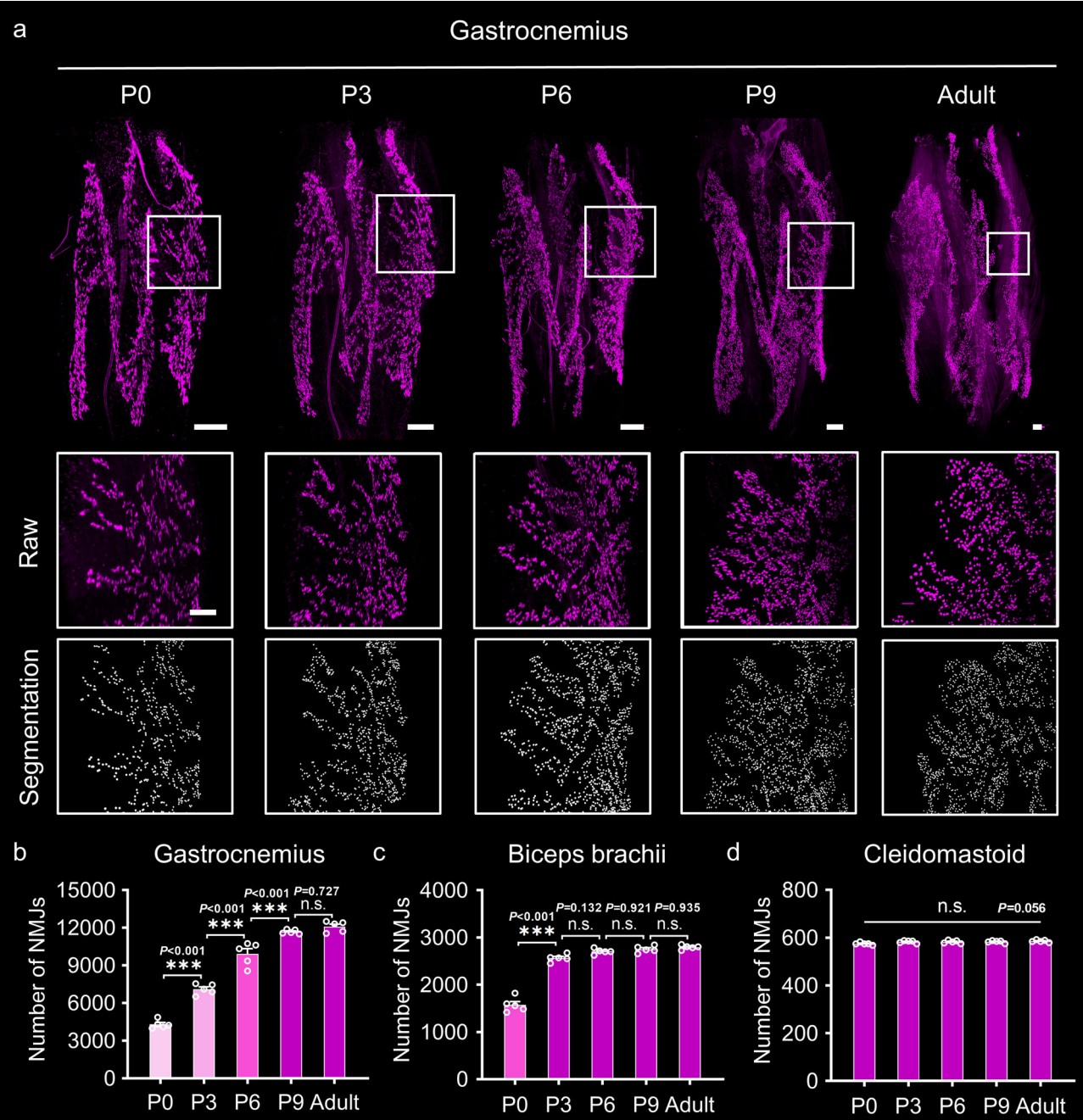

**Fig. 6 3D analysis of NMJs during postnatal development. a** 3D distributions of NMJs in the gastrocnemius at different time points (P0, P3, P6, P9, and adulthood); scale bar: 200 μm. The images in the white frames are enlarged and shown in the second row; scale bar: 100 μm. The images in the third row show the segmentation of NMJs. **b–d** The quantification of NMJs in the gastrocnemius (**b**), biceps brachii (**c**), and cleidomastoid (**d**) at different time points (*n* = 5 independent animals for each muscle at each time point). All values are presented as the means ± SEM; statistical significance in (b-d) (n.s. represents not significant and ***$P < 0.001$) was assessed using one-way ANOVA followed by Tukey post hoc test.

Intramuscular motor innervation is an essential process during neuromuscular development. Here, we focused on collagen XXV, transmembrane-type collagen, identified initially as a component of the senile plaque, amyloid of Alzheimer's disease of the brain[42]. Tanaka et al. demonstrated that without collagen XXV, motor axons successfully reach target muscles but fail to arborize to intramuscular branches[43]. Moreover, muscle-derived collagen XXV mRNA is highly expressed in developing muscles, decreases with muscle development, and disappears almost entirely in adulthood[31,43]. Hence, the extent of its expression constitutes a suitable parameter that should reflect the development of the motor nerves quantitatively.

Clinically, the distribution of intramuscular nerves can be established by anatomical microdissection or using Sihler's staining technique, which provides references for nerve graft and intramuscular injection[44,45]. In experimental research, quantitatively analyzing the detailed characteristics of the distribution of nerves is a step that must be performed to determine changes in intramuscular nerves during development, injury, or regeneration. However, due to limited molecular and optical penetration in large opaque tissues, quantitative analyses of nerve branches

are primarily performed in flat muscles, such as the diaphragm and the *latissimus dorsi*[46,47]. Thanks to advances in tissue optical clearing, nerve distributions in multi-size skeletal muscles can now be 3D-visualized, which facilitates the examination of intramuscular nerves at the overall level of the muscle during postnatal development.

Previous studies seldom reported changes in NMJ numbers because of the difficulty of analyzing total NMJs in whole muscles using traditional histological methods based on the sparse distribution of NMJs in opaque muscle tissues. Tapia et al. found NMJ numbers in the cleidomastoid to be stable from birth[22], which is consistent with our result. However, they detected about 410 NMJs, which is lower than ours. This inconsistency may result from the difference in research methods: Tapia et al. used the confocal microscope to quantify NMJs in opaque muscles and reconstructed the data in 2-dimensions by creating a maximum intensity projection for each stack, with the overlap of NMJ signals at depth possibly leading to the underestimation of NMJ numbers: we used the light-sheet microscope to detect NMJs in transparent muscles with the aid of tissue optical clearing and reconstructed the data in 3-dimensions, which helps to detect all NMJs for precise counting. During the quantification of the number of NMJs, the fluorescent signal of the postsynaptic AChR cluster was identified as a target for counting, probably resulting in the overestimation of NMJ numbers due to the potential existence of ectopic aneural AChR clusters during NMJ formation. Therefore, to ensure accurate counting, AChR cluster signals that did not overlap with the YFP signals of motor terminals, i.e., non-innervating AChR clusters, were excluded by manual correction.

Each myofiber is innervated by one single NMJ, whose numbers should correspond to myofiber numbers in the normal muscle. But in this study, NMJ numbers were not always consistent with myofiber numbers at the same time point in the same muscle. For example, at the stage of NMJ increase, NMJ numbers were slightly fewer than myofiber numbers. This inconsistency might be the result of myofiber formation beginning earlier than NMJ formation. However, given that newly formed myotubes are innervated immediately by the axonal terminal to form NMJ[19,48], we speculated that the data bias generated by different quantitative methods for NMJs in 3D and myofibers in 2D sections possibly provides a reason for this inconsistency. The gastrocnemius in P9 and adult mice had more NMJ numbers (P9: $11676 \pm 149$; adult: $12089 \pm 175$) than myofiber numbers (P9: $10296 \pm 314$; adult: $10345 \pm 297$). Generally, during NMJ formation, many small and primitive AChR clusters form on myotubes (muscle pre-patterning)[49]. During NMJ number quantification via AChR cluster counting, ectopic aneural AChR clusters may lead to the overestimation of NMJ numbers. However, NMJ formation in the gastrocnemius of P9 and adult mice was over, and aneural AChR clusters had disappeared[50], and this did not affect the accurate quantification of NMJ number. Thus, we speculated that the major reason for the inconsistency between NMJ and myofiber numbers in the gastrocnemius of P9 and adult mice is the underestimation of myofiber number due to the special anatomy of the gastrocnemius. Some myofibers in certain compartments of the gastrocnemius are pinnately distributed[51], with myofiber numbers in the transverse section of the muscle belly fewer than the total number in the whole muscle.

Alongside behavioral performances during postnatal development, the findings on myofiber and neural structure development changes in mouse muscles can help to better understand that motor development requires the growth of an infant's body for support[52]. At birth (P0), the motor performance of neonatal mice was dominated by cervical function, with the limbs almost immobile. During this time, myofiber and NMJ formations in the

cervical cleidomastoid were complete, with their numbers both reaching adult-level figures, which is consistent with the results in another study[22]. And the numbers in limb muscles were much lower than those in adult mice. During subsequent development, when the forelimbs or hindlimbs dominate motor behaviors, NMJ and myofiber numbers in the skeletal muscles of corresponding body parts both developed to adult levels. These results demonstrate that the improvement in motor function after birth is closely associated with the increase in NMJ and myofiber numbers in skeletal muscles, which correlates structural data with functional performance.

Knowledge of NMJ formation during postnatal development should boost future investigations on synaptic development. The mechanism of synapse elimination has been studied extensively[53,54], with some studies showing that axon loss resulting from synapse elimination in the central nervous system is associated with an increase in synapse numbers[55–57]. However, in the peripheral nervous system, there is no evidence of whether an increase in NMJ numbers is associated with axon loss during synapse elimination[22]. This may be because previous studies extensively used cervical muscles, such as the sternomastoid and cleidomastoid, to investigate synapse elimination during postnatal development[21,22,58]; NMJ numbers in these muscles do not increase after birth, making it impossible to determine the relationship between NMJ count rise and axon loss. In this study, NMJ formation in mouse limb muscles continued and was possibly not completed in hindlimb muscles until P9, which provides references for future investigations on NMJ development. One recent study found that Bessel light-sheet microscopy with content-aware compressed sensing (CACS) computation pipeline generated 3D images with high, isotropic spatial resolution, enabling not only the acquisition of the spatial distribution characteristics of NMJs at the overall level of the skeletal muscles but also the discernment of the fine structure of single NMJs, such as pre-synaptic and post-synaptic morphologies[27]. Hence, future research should use CACS Bessel light-sheet microscopy to target NMJs in hindlimb muscles for possible observation of both NMJ formation and axon loss during postnatal development, which would contribute to the exploration of the relationship between synaptogenesis and synapse elimination.

In conclusion, we examined the 3D structural features of the neuromuscular system in the skeletal muscles of mice during postnatal motor development by combining tissue optical clearing with optical imaging techniques. We present evidence of myofiber hyperplasia, axonal arborization, and NMJ formation in postnatal muscles. The differences in these structural features in various muscles at the same time point after birth demonstrate the developmental heterochrony of the neuromuscular system in different body parts. Cervical muscles are the most developed, followed by forelimb muscles; hindlimb muscles are the least developed muscles during early postnatal development, which is consistent with the principle of motor development. Our data provide clarity on the gaps in the 3D structural information on the neuromuscular system during motor development and deepen the understanding of the structure and function from a 3D perspective.

## Methods

**Animals**. Transgenic mice were bred and housed in accordance with the Experimental Animal Management Ordinance of Hubei Province, P. R. China, and the guidelines from the Huazhong University of Science and Technology and were approved by the Institutional Animal Ethics Committee of Huazhong University of Science and Technology. Transgenic *Thy1*-YFP-16-line mice obtained from Jackson Laboratory (Bar Harbor, Maine, USA) were bred to produce timed litters (embryonic mice and postnatal pups). Animals were housed in a specific pathogen-free animal environment under a 12/12-hour light/dark cycle and provided with food and water *ad libitum*. To get the exact gestational age of embryonic mice, the

morning on which the copulation plug became apparent was designated as the first-day post-*coitum* (or E0.5). For the postnatal pups, pregnant females were isolated and checked daily. The data at birth were designated postnatal day 0 (P0).

**Assessment of motor development in neonatal mice**. Assessment methods used in this study were performed as reported previously[9,59]. Before testing, the mothers and pups were transferred from the housing cages to the testing room for 2 h for habituation to the testing conditions. The pups were then removed and placed on the heating pad, which maintains the pups' temperature because they are poikilothermic for a few days after birth. Gloves were not allowed throughout the procedure; therefore, to avoid meddling with the typical odor situation, experimenters washed their hands carefully and rubbed them with sawdust from the corresponding pups' cages before touching the said pups. The above operation was repeated for tests with pups from each cage.

*Righting response*. Each pup was placed gently on its back on a platform and held for several seconds by the fingers of an investigator, and the time taken from when the fingers were pulled back to when the pup flipped over onto its belly with four paws touching the plate was recorded. If a pup failed to turn over within 20 s, the time was recorded as 20 s. Each pup took this test three times, with a 10 min rest imposed between tests.

*Limb placement*. Each pup was handled between the thumb and forefinger gently around the trunk, making sure not to limit the rotation of the trunk and the movement of the forelimb or hindlimb. A thin metal bar was then used to touch the dorsum of the forepaw or hindpaw, and a scoring system was employed to determine the pup's response. Score 1: the pup raises its limb or retracts its paw when the metal bar touches its paw. Score 2: the pup raises its limb and places it on the metal bar. To reduce the pup's response contingency, the same limb was tested three times. Therefore, minimum and maximum total scores for a pup would be 3 and 6, respectively, with a total score of 3 indicating no response; a total score of 4, occasional appearance; a total score of 5, frequent appearance; a total score of 6, stable appearance.

*Adult walking pattern*. Each pup was placed on a platform flat on its stomach, and its pattern was scored. The score was used to evaluate the walking pattern without interference from external conditions. Score 1: the body is mostly immobile and is poorly supported by its limbs. Score 2: pivoting movement – the pup makes broad swipes with the forepaws to generate a paddle-like movement, and its hindlimbs are motionless. Score 3: crawling movement – the pup begins to crawl forward with a stagger depending on the paddling movement of the paws. The hindlimbs often fail to keep up with the forelimbs. Score 4: walking movement – the pup puts all four paws on the ground and develops a linear movement with a straight walk.

*Cliff drop avoidance*. Each pup was placed on the edge of a platform (about 2 centimeters in height for safety) with its nose and forefeet over the edge, and its response was scored. The position of the forepaws is crucial to the accurate measurement of a pup's response to the cliff: the edge of the platform is under the junction of the ulna and the metatarsal bones. The resulting scores are used to define the behavioral characteristics of the pup. Score 1: the pup remains static or falls from the edge. Score 2: the pup turns its head aside to avoid the cliff. Score 3: the pup turns its head aside, pulls back the forepaws from the edge, and puts them on the platform. Score 4: the pup retracts directly from the edge by backing up.

**Labeling NMJs in skeletal muscles**. For adult mice, Alexa Fluor 647-conjugated α-bungarotoxin (α-BTX 647, Invitrogen, Carlsbad, CA, USA) was injected (0.3 μg/g) via the intravenous tail and a conjugation time of 2 h prior to perfusion[25]. For newborn and juvenile mice, an inner canthus veniplex injection was used in place of the intravenous tail injection utilized in adults because the tail vein at this stage of development is too thin to be operated upon. After injection, the pups were placed on the heating pad to maintain their temperature, and enough time (30 min) was allocated for conjugation.

**Preparation of mouse skeletal muscles**. In this study, common experimental muscles from different body parts, including the cleidomastoid, biceps brachii, gastrocnemius, and tibialis anterior were selected. The cleidomastoid is located in the cervical part and controls the head movement. The biceps brachii is situated in the forelimb and enables the elbow and shoulder joints to perform forearm supination. And the gastrocnemius and tibialis anterior both reside in the hindlimb and are essential for walking. Animals in each age group were randomly selected.

For tissue preparation for mRNA analysis, postnatal and adult mice were deeply anesthetized with a mixture of 10% urethane, 2% chloral hydrate (8 mL/kg), and 0.3% xylazine (5 mL/kg). To ensure that the neonatal muscles were adequate for RNA extraction, the muscles from the cervical region (sternomastoid, cleidomastoid, and trapezius), forelimb (biceps brachii and triceps brachii), and hindlimb (gastrocnemius, tibialis anterior, and soleus) were dissected quickly, washed with pre-cooling 0.01 M phosphate-buffered saline (PBS, Sigma-Aldrich, St. Louis, USA), and stored immediately in RNAstore Reagent (DP408, Tiangen Biotech Co., Ltd, Beijing, China) at 4 °C. For embryonic tissues, pregnant mice

were deeply anesthetized at around midday on days 14 and 18 post-coitum, that is, at gestational ages E13.5 and E17.5. Their embryos were then rapidly removed from the uterus one after the other and washed with pre-cooling 0.01 M PBS, and whole hindlimbs were dissected swiftly and stored immediately in RNAstore Reagent at 4 °C.

For fluorescence imaging, mice were perfused intracardially with 0.01 M PBS, then 4% paraformaldehyde (PFA, Sigma-Aldrich, St. Louis, USA) in PBS. The perfusion volume was selected based on the sizes of the mice. 50 mL PBS and 30 mL PFA were used for adult mice, while 5 mL PBS and 3 mL PFA were used for neonatal mice. After perfusion, the cleidomastoid (cervical region), biceps brachii (forelimb), gastrocnemius (hindlimb), and tibialis anterior (hindlimb) were dissected carefully under a stereomicroscope (SZ61, Olympus, Tokyo, Japan). All harvested muscles were post-fixed in 4% PFA overnight at 4 °C and rinsed several times with PBS the following day before clearing or sectioning. For muscle section collection, muscles were first embedded in 3% agarose and subsequently sliced into 100 μm thick transverse sections using a vibratome (Leica VT1200S, Wentzler, Germany).

**Optical clearing of skeletal muscles**. FDISCO (three-dimensional imaging of solvent-cleared organs with superior fluorescence preserving capability) was performed as described in the original literature[41]. And for the muscles in neonatal mice, the dehydration time was adjusted according to muscle sizes. Before clearing, tetrahydrofuran (THF, Sigma-Aldrich, St. Louis, MO, USA) and dibenzyl ether (DBE, Sigma-Aldrich, St. Louis, MO, USA) were pre-processed with basic activated aluminum oxide (Sinopharm Chemical Reagent Co., Ltd, Shanghai, China) using column absorption chromatography to remove peroxides. The pH of THF solutions (mixed with $dH_2O$) at various concentrations (50, 70, 80, and 100 vol%) were adjusted to about 9.0 with trimethylamine. Dehydration time was set depending on muscle sizes: for P0 and P3 muscles, each step took only 5 minutes; for P6 and P9 muscles, each step took 10 and 20 minutes, respectively; and for adult muscles, each step took at least 2.5 hours. After dehydration, pure DBE (Sigma-Aldrich, St. Louis, MO, USA) was used to clear the muscles as a refractive index matching solution. All the steps were performed at 4 °C with slight shaking.

**Immunostaining**. The following antibodies were used in this procedure: primary antibodies – anti-Dystrophin (1:200 dilution; ab15277, Abcam, Cambridge, MA, USA) and anti-Mki67 (1:500 dilution; K009725P, Solarbio®, Beijing, China); secondary antibody – Alexa Fluor 555 goat anti-rabbit IgG (H + L) (1:500 dilution; A21429, Invitrogen, Carlsbad, CA, USA).

100 μm thick muscle slices from the transverse section of the muscle belly were blocked in PBS/0.2% Triton X-100 (Sigma-Aldrich, St. Louis, MO, USA)/10% goat serum at room temperature for 30 minutes, incubated with primary antibody dilutions in PBS/0.2% Triton X-100/5% goat serum with slight shaking overnight at 4 °C, washed with PBS/0.2% Triton X-100 several times at room temperature, incubated with the secondary antibody diluted in PBS/0.2% Triton X-100/5% goat serum with slight shaking for 2 hours at room temperature, washed with PBS/0.2% Triton X-100 several times, and imaged.

For nuclei staining, 100 μm thick muscle slices from the transverse section were incubated with DAPI (4',6-Diamidino-2-Phenylindole, 10 μg/mL; D1306, Invitrogen, Carlsbad, CA, USA) in PBS/0.2% Triton X-100 for 2 hours at room temperature, with gentle oscillation, and washed with PBS.

**Fluorescence microscopy**

*Confocal microscopy*. Muscle slices were mounted on two cover glasses and imaged with an inverted confocal fluorescence microscope (LSM710, Zeiss, Oberkochen, Germany) equipped with the Fluar 10×/0.5 objective (dry; working distance, 2.0 mm), Plan-Apochromat 20×/0.8 objective (dry; working distance, 0.55 mm), Plan-Apochromat 40×/1.4 objective (oil; working distance, 0.13 mm), and alpha Plan-Apochromat ×63/1.46 objective (oil; working distance, 0.10 mm).

*Light-sheet fluorescence microscopy*. Transparent muscles were imaged with the light-sheet fluorescence microscope (LSFM, UltraMicroscope, LaVision BioTec, Germany) equipped with an sCMOS camera (Andor Neo), a 2×/0.5 objective lens furnished with a dipping cap, and an Olympus MVX10 zoom microscope body (magnification range of ×0.63 to ×6.3). For NMJs labeled with α-BTX 647 and nerve branches with YFP, 633 nm and 488 nm were applied, respectively, as the exciting wavelengths. The z step size was set to 3 μm for neonatal muscles and 5 μm for adult muscles. Images of the samples were acquired for subsequent processing and analysis after the appropriate setting of the imaging parameters.

**Quantitative reverse transcriptase-PCR analyses of muscles**. Total RNA was extracted from the neck, forelimb, and hindlimb muscles of mice at different times using the RNAprep Pure Tissue Kit (DP431, Tiangen Biotech Co., Ltd, Beijing, China) and converted to cDNA utilizing the HiFiScript cDNA Synthesis Kit (CW2569, CoWin Biosciences, China). Because an individual muscle in a neonatal mouse is too small to yield enough RNA, multiple muscles from the same body part, such as the sternomastoid, cleidomastoid, and trapezius in the cervical part, were dissected for the extraction of sufficient RNA, ensuring that the obtained results more representative. Quantitative real-time PCR was performed using the

ChamQTM Universal SYBR® qPCR Master Mix (Q711-02, Vazyme, China) and analyzed with the QuantStudioTM Design & Analysis Software (Thermo Fisher, Waltham, MA, USA). Each sample was scrutinized three times, and the average finding was taken to reduce experimental errors. Expression values were normalized to GAPDH mRNA. The primer sequences used for PCR were as follows: 5-TCCTTCCATCCGCTGTCT-3 and 5-TCCCTGGCCGTTCTTATT-3 for CLAC-P/collagen XXV and 5-AACGACCCCTTCATTGAC-3 and 5-GAAGACACCAGTAGACTCCAC-3 for GAPDH.

**Processing of imaging data**. Zen 2011 SP2 (Version 8.0.0.273, Zeiss, Oberkochen, Germany) and ImSpector (Version 4.0.360, LaVision BioTec, Germany) were used for data collection. The obtained images were processed and analyzed with ImageJ (Version 1.51j8, NIH, Bethesda, MD, USA), Imaris (Version 7.2.3, Bitplane, Zurich, Switzerland), and MATLAB (Version 2014a, Mathworks™, Natick, MA, USA).

**Quantification of KI67 positive cells and myofiber numbers**. Stained transverse muscle sections were imaged using an inverted confocal fluorescence microscope. The obtained images were viewed and processed with ImageJ.

KI67-positive nuclei and DAPI-labeled nuclei were extracted using the threshold function, their numbers were measured using the "analyze particles" function, and the percentages of KI67-positive cells were calculated. The mean percentage of KI67-positive cells in each muscle slice was obtained by averaging the values of three regions.

For the small-volume cleidomastoid, myofibers were counted manually. For other larger muscles (biceps brachii, gastrocnemius, and tibialis anterior), the calculation procedure is shown in Supplementary Fig. 7. The "Rectangle" tool was used to crop several regions with equal sizes as unit regions, and the fiber numbers in these unit regions were counted and the average further calculated. The "Freehand Selection" tool was then employed to determine the edges of entire muscle sections, and the total areas were measured. Subsequently, the number of unit regions contained in any given entire muscle section was established and multiplied by the average fiber number per unit region to obtain the total number of that whole muscle section.

**Quantification of number of myofibers with central myonuclei**. First, the total number of myofibers in the muscle section was counted based on the staining of myofiber membranes. Then the myofibers without myonuclei were identified and counted and excluded from total myofiber number, thereby obtaining the number of myofibers with myonuclei. Finally, the myofibers with central myonuclei were identified and counted, and their proportion to the number of myofibers with myonuclei was calculated as the percentage of myofiber with central myonuclei.

**Quantification of the relative range of the innervation zone**. The relative range of the innervation zone was expressed as a percentage of the width of the entire muscle. Based on the 3D imaging data of intramuscular nerves and NMJs, the maximum intensity projection of the z stacks was established using ImageJ. This was accomplished by drawing two tangents to the muscle fibers on either side of the innervation zone (see white lines in Supplementary Fig. 2b). The orthogonal distance between these lines (yellow arrows) was defined as the innervation zone width, and the muscle width was projected onto this same axis (white arrows). The ratio of these two lengths was conveyed as a percentage.

**Quantification of the intramuscular innervation pattern**. Nerve branches were traced manually using the AutoDepth method of the Filament module in the Imaris. We defined primary branches as originating from nerve branches that entered the muscles and extended from proximal to distal and secondary branches as originating from primary branches and ending at NMJs. Primary and secondary branch numbers were counted to reflect the developmental level of the intramuscular branching pattern. The complete process is shown in Supplementary Fig. 8. First, the Filament module in the toolbar was selected, and the automatic creation was skipped for the manual editing mode (Supplementary Fig. 8a). Next, the Cone style was selected, and the AutoDepth method was used (Supplementary Fig. 8b, c). "AutoDepth" can recognize the signal in 3D space automatically when drawing a cone from the starting point to the endpoint. The signal depth is automatically calculated and recognized to locate the signal and trace continuous nerve fibers.

**Measurement of NMJ numbers**. The entire process is shown in Supplementary Fig. 9. First, the Spots module was selected in the toolbar, and the calculation mode was scrutinized for the region of interest (Supplementary Fig. 9a). Then the size of the calculated region was set based on coordinates (Supplementary Fig. 9b), and the approximate diameter of the NMJ particle was entered (Supplementary Fig. 9c). Next, the algorithm "Intensity StdDev" was selected to identify the particle, and the number of identified particles was determined by adjusting the threshold range (Supplementary Fig. 9d); parameter values are obtainable based on image quality. Finally, the segmentation process was completed, and the "Add/Delete" mode was selected manually to correct the

segmentation results (Supplementary Fig. 9e). With the use of artificial judgment, the unrecognized signal points were added, and incorrect data points from aneural AChR clusters, residual bubbles in samples, and the autofluorescence of muscle fibers and blood vessels were eliminated.

**Statistical and reproducibility**. Statistical analysis was performed using the SPSS software. Sample sizes (n) are presented in the figure legends, with animals randomly assigned to groups for experiments. And n value was defined as the number of independent animals in this paper. Data are presented as the means ± SEM. The Shapiro-Wilk test, a method for small sample sizes, was used to assess the normality of data distribution in each experiment. The heterogeneity of variance was evaluated using the Levene test, a stable method for both normally and non-normally distributed data. P values were calculated using one-way ANOVA (Figs. 2c, g–i, 3e–g, 4b, 5j–l, and 6b–d, and Supplementary Figs. 1c, 2c, and 6d) and two-way ANOVA (Fig. 2e and Fig. 4c). If the variance was homogeneous, we used the Tukey post hoc test for multiple comparisons, otherwise, we used Dunnett's T3 post hoc test for multiple comparisons. In this study, $P < 0.05$ was considered significant (*$P < 0.05$, **$P < 0.01$, and ***$P < 0.001$).

**Reporting summary**. Further information on research design is available in the Nature Research Reporting Summary linked to this article.

## Data availability

The datasets generated during and/or analyzed during the current study are available from the corresponding author on reasonable request. All source data for the graphs and charts are in Supplementary Data 1.

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

## Acknowledgements

This study was supported by the National Key Research and Development Program of China (Grant No. 2017YFA0700501), the National Natural Science Foundation of China (Grant Nos. 61860206009, 81870934, 62105113, 81961138015, and 81701354), Innovation Project of Optics Valley Laboratory (Grant No. OVL2021BG011), and the Innovation Fund of WNLO. The authors also thank Baoguo Jiang from Peking University People's Hospital for providing the *Thy1*-YFP-16 transgenic mice, Bin Zhang from Tongji Medical College of Huazhong University of Science and Technology for helpful suggestions, and the Optical Bio-imaging Core Facility of WNLO-HUST for support with data acquisition.

## Author contributions

D.Z. and T.Y. conceived and designed the study. J.X. and J.Z. performed tissue clearing and imaging. J.X. and Y.Y. performed the qRT-PCR and the assessment of the motor performance in neonatal mice. J.X., Y.L., and A.X. performed image processing and analysis. J.X. and J.Z. wrote the paper. D.Z. and T.Y. supervised the project and revised the manuscript. J.Z. and D.L. gave valuable comments and suggestions for this study. All authors read and approved the final manuscript.

## Competing interests

The authors declare no competing interests.
