## [Peer Review File · Communications Biology]

Reviewers' comments:

Reviewer #1 (Remarks to the Author):

In this clear, logical and well-presented manuscript, Xu and colleagues methodically assess the neuromuscular system, using exquisite clearing techniques, across early postnatal development in mice. By comparing motor function, myofiber numbers, motor nerve branching and neuromuscular junction populations across muscles and postnatal development, the authors provide compelling morphological data supporting the heterochronic development of the neuromuscular system. There is excellent consistency across analyses, bolstering confidence in their conclusion. Overall, the manuscript is worthy of publication, and need not be changed. I really appreciated the careful assessment of the four different muscles across timepoints and applaud the authors for such a concise and clear description of the experiments. Nevertheless, I have a few minor points that the authors may wish to address.

- The images presented in Figure 2B present a clear picture, which is supported by the analyses in subsequent panels of the figure; however, quantification of the phenotype in 2B would improve the figure.
- Red/green colour combinations should be avoided for colour-blind readers (e.g. Fig. 3 and 4).
- It would be really interesting to assess the percentage of polyinnervated NMJs (as a readout of synapse elimination) across muscles and ages to see whether this process of synaptic maturation correlates with the data presented in the final figure. The authors may be referring to this on lines 397-398, when they state, "...the consistency of NMJ numbers in the cervical muscles after birth makes it impossible to determine the relationship between the NMJ increase and axon loss." However, it is unclear to me why this is the case – some clarity of this would be appreciated.
- The information provided on lines 339-342 in the Discussion would perhaps be better placed/more useful if moved to the Results section, where the data are first reported.

Reviewer #2 (Remarks to the Author):

The manuscript "Three-Dimensional Mapping Reveals Heterochronic Development of the Neuromuscular System in Postnatal Skeletal Muscles" is a very interesting manuscript, having as its main aim to "investigate the structural development of the neuromuscular system in different muscles from a novel perspective". The authors claim that they demonstrate the heterochrony of skeletal muscle development "in different body parts and correlate it with motor development". The manuscript contains a very interesting methodological approach and results that, in my opinion, deserve a better manuscript not only in technical and formal terms but also in the way the results are described and discussed.

++ Major comments:

1.- My main conceptual concern is that, despite the interesting 3D structural information on the neuromuscular innervation of developing muscles, the author's enthusiasm regarding the novelty of this study about the heterochrony of skeletal muscle development seems not to be strongly supported, particularly by the way the text is written. In this regard:

. Although specific muscles studied throughout this manuscript were selected according to their distribution within the A-P axis, the contribution of other important parameters, such as (i) their fast/slow fiber content, (ii) their distance from the spinal cord, or (iii) the size of their motor units, to mention some, have not been pondered or properly discussed.

. Related to this, the entire discussion section is rather poor. In my opinion, this section needs a better conceptualization of the data regarding the literature. Indeed, some important points are discussed in

large and highly speculative paragraphs, repeating results, and with no references (see, for instance, lines 375-390). Another example is the paragraph addressing the inconsistency between myofiber v/s NMJ numbers (lines 368-374), a point for which there is abundant literature to be discussed. This includes, on one hand, the virtually immediate innervation of recently formed embryonic skeletal muscle fibers and, on the other hand, the potential existence of ectopic aggregates of AChRs.

. Also, the conceptual contribution of the experiments aimed at determining cell proliferation (line 148 onwards) seems not accurately presented or discussed. Although I agree with the idea that these experiments "should provide a quantifiable parameter for the determination of muscle development" it is not clear from the manuscript how this data could contribute to discriminating between muscle fiber hypertrophy and hyperplasia.

. The article will benefit from the quantification of central nuclei in Fig. 2B.

2.- As the methodology is one of the key contributions of this manuscript, some of the novel methods require a much more detailed description. See, for instance, the "Quantification of the intramuscular innervation pattern" and the "Measurement of NMJ numbers" descriptions.

3.- To help a better data conceptualization, please avoid purely methodological descriptions in the discussion (see, for instance, lines 339-342). Also, specific figures should not be cited in this section.

++ My next comments are as follows:

+ General:

. Although the manuscript is understandable, English correction is highly recommended, not only to correct many mistakes but mainly to improve the reading fluency of the manuscript.

. Please reconsider the use of a black background in all figures as it does not seem to be always helpful.

. When addressing the number of NMJs, it is important to mention that these quantifications are based mainly on postsynaptic AChR aggregates and to discuss the potential limitations of this choice.

+ Specific:

. There is no description for Fig. 2A regarding the muscle pictures.

. Please include statistical analyses of the data in Fig. 4B

. Please be more specific in the description of the usefulness of addressing Collagen XXV levels in the context of this manuscript.

. Please briefly explain the selection of the Shapiro-Wilk and Levene tests for the statistical analyses.

Rebuttal Letter

Dear Reviewers,

We would like to thank you for the valuable comments and suggestions that help us further improving this manuscript. Now we will respond to the comments one by one, and show the changes in the following pages and also use the highlighting method to mark the changes in the revised manuscript.

We look forward to hearing from you at your earliest convenience.

Yours sincerely,

Dan Zhu, PhD, SPIE Fellow,

Professor, Britton Chance Center for Biomedical Photonics, Huazhong University of Science and Technology

Deputy Director, Wuhan National Laboratory for Optoelectronics

Secretary General & Vice Chairman, Biomedical Photonics Committee of Chinese Optical Society.

1037 Luoyu Road, Wuhan 430074, P.R. China

E-mail: dawnzh@mail.hust.edu.cn

For the comments of the Reviewer #1

In this clear, logical and well-presented manuscript, Xu and colleagues methodically assess the neuromuscular system, using exquisite clearing techniques, across early postnatal development in mice. By comparing motor function, myofiber numbers, motor nerve branching and neuromuscular junction populations across muscles and postnatal development, the authors provide compelling morphological data supporting the heterochronic development of the neuromuscular system. There is excellent consistency across analyses, bolstering confidence in their conclusion. Overall, the manuscript is worthy of publication, and need not be changed. I really appreciated the careful assessment of the four different muscles across timepoints and applaud the authors for such a concise and clear description of the experiments. Nevertheless, I have a few minor points that the authors may wish to address.

1. The images presented in Figure 2B present a clear picture, which is supported by the analyses in subsequent panels of the figure; however, quantification of the phenotype in 2B would improve the figure.

Response: Thank you for your suggestion. According to your suggestion, we analyzed the percentage of the myofibers with central nuclei in different muscles during postnatal development, and added quantitative result in Figure 2C and some descriptions.

Figure 2. Postnatal development of myofibers in different skeletal muscles.

Line 150 to 157: According to findings from the quantification of the percentage of myofibers with central nuclei (**Figure 2C**), central myonuclei rarely occupied the cleidomastoid areas after birth (4.60%), with most migrating to the periphery, pointing to the completion of myogenesis. As for the biceps brachii and gastrocnemius on P0, the percentages of myofibers with central myonuclei were 19.57% and 34.49%, respectively, indicating that some fibers in these two muscles were still immature myotubes. Subsequently, the central nuclei almost disappeared in the biceps brachii on P3 (3.93%) and in the gastrocnemius on P6 (2.38%), showing that myotubes in the biceps brachii mature before those in the gastrocnemius.

2. *Red/green colour combinations should be avoided for colour-blind readers (e.g. Fig. 3 and 4).*

Response: Thank you for your suggestion. To avoid red/green color combinations, we have modified the pseudo-colors in **Figure 3**, **Figure 4**, **Figure 6**, **Figure S2**, and **Figure S6**.

3. *It would be really interesting to assess the percentage of polyinnervated NMJs (as a readout of synapse elimination) across muscles and ages to see whether this process of synaptic maturation correlates with the data presented in the final figure. The authors may be referring to this on lines 397-398, when they state, "...the consistency of NMJ numbers in the cervical muscles after birth makes it impossible to determine the relationship between the NMJ increase and axon loss." However, it is unclear to me why this is the case – some clarity of this would be appreciated.*

Response: Thank you for your suggestions. For the investigation of the relationship between synapse elimination and synapse number, it is required to detect all NMJs in whole skeletal muscle while obtaining the fine structure of individual NMJs at high resolution. Recent development in optical imaging techniques will potentially meet this requirement. We added some discussions and also modified some descriptions in the manuscript.

Line 444 to 452: One recent study found that Bessel light-sheet microscopy with content-aware compressed sensing (CACS) computation pipeline generated 3D images with high, isotropic spatial resolution, enabling not only the acquisition of the spatial distribution characteristics of NMJs at the overall level of the skeletal muscles but also the discernment of the fine structure of single NMJs, such as pre-synaptic and post-synaptic morphologies²⁷. Hence, future research should use CACS Bessel sheet microscopy to target NMJs in hindlimb muscles for possible observation of both NMJ formation and axon loss during postnatal development, which would contribute to the exploration of the relationship between synaptogenesis and synapse elimination.

Line 437 to 442: However, in the peripheral nervous system, there is no evidence of whether an increase in NMJ numbers is associated with axon loss during synapse elimination²². This may be because previous studies extensively used cervical muscles, such as the sternomastoid and cleidomastoid, to investigate synapse elimination during postnatal development^{21,22,59}; NMJ numbers in these muscles do not increase after birth, making it impossible to determine the relationship between NMJ count rise and axon loss.

4. *The information provided on lines 339-342 in the Discussion would perhaps be better placed/more useful if moved to the Results section, where the data are first reported.*

Response: Thank you for your suggestion. We moved this part to the **Results** section.

Line 238 to 241: Because an individual muscle in a neonatal mouse is too small to yield enough RNA, multiple muscles from the same body part, such as the sternomastoid, cleidomastoid, and trapezius in the cervical part, were dissected for the extraction of sufficient RNA, ensuring that the obtained results more representative.

For the comments of the Reviewer #2

The manuscript “Three-Dimensional Mapping Reveals Heterochronic Development of the Neuromuscular System in Postnatal Skeletal Muscles” is a very interesting manuscript, having as its main aim to “investigate the structural development of the neuromuscular system in different muscles from a novel perspective”. The authors claim that they demonstrate the heterochrony of skeletal muscle development “in different body parts and correlate it with motor development”. The manuscript contains a very interesting methodological approach and results that, in my opinion, deserve a better manuscript not only in technical and formal terms but also in the way the results are described and discussed.

Major comments:

1. My main conceptual concern is that, despite the interesting 3D structural information on the neuromuscular innervation of developing muscles, the author’s enthusiasm regarding the novelty of this study about the heterochrony of skeletal muscle development seems not to be strongly supported, particularly by the way the text is written. In this regard:

a) Although specific muscles studied throughout this manuscript were selected according to their distribution within the A-P axis, the contribution of other important parameters, such as (i) their fast/slow fiber content, (ii) their distance from the spinal cord, or (iii) the size of their motor units, to mention some, have not been pondered or properly discussed.

Response: Thank you for your suggestion. According to your suggestion, we added some discussion on the selection of experimental muscles.

Line 314 to 319: Specific skeletal muscles were selected in this study according to their distribution in the anteroposterior (A-P) axis, including the cleidomastoid, biceps brachii, gastrocnemius, and tibialis anterior, which are all ‘fast’ muscles dominated by type II fibers³²⁻³⁴. ‘Slow’ muscle types were not examined in this investigation. In addition, other important parameters, such as the distance between the muscle and the spinal cord (proximodistal axis)³⁵, contribute to developmental characteristics and deserve to be scrutinized in future work.

b) Related to this, the entire discussion section is rather poor. In my opinion, this section needs a better conceptualization of the data regarding the literature. Indeed, some important points are discussed in large and highly speculative paragraphs, repeating results, and with no references (see, for instance, lines 375-390).

Response: Thank you for the suggestion. According to your suggestion, we modified the discussion in this manuscript.

Line 420 to 433: Alongside behavioral performances during postnatal development, the

findings on myofiber and neural structure development changes in mouse muscles can help to better understand that motor development requires the growth of an infant's body for support⁵³. At birth (P0), the motor performance of neonatal mice was dominated by cervical function, with the limbs almost immobile. During this time, myofiber and NMJ formations in the cervical cleidomastoid were complete, with their numbers both reaching adult-level figures, which is consistent with the results in another study²². And the numbers in limb muscles were much lower than those in adult mice. During subsequent development, when the forelimbs or hindlimbs dominate motor behaviors, NMJ and myofiber numbers in the skeletal muscles of corresponding body parts both developed to adult levels. These results demonstrate that the improvement in motor function after birth is closely associated with the increase in NMJ and myofiber numbers in skeletal muscles, which correlates structural data with functional performance. In addition, CNS maturation is generally believed to be the main driving force behind motor development³. Consequently, the developmental relationships between CNS, body hardware and motor function are worth exploring further.

c) *Another example is the paragraph addressing the inconsistency between myofiber v/s NMJ numbers (lines 368-374), a point for which there is abundant literature to be discussed. This includes, on one hand, the virtually immediate innervation of recently formed embryonic skeletal muscle fibers and, on the other hand, the potential existence of ectopic aggregates of AChRs.*

Response: Thank you for the suggestion. According to your suggestion, we enriched the discussion on the inconsistency between myofiber and NMJ numbers.

Line 401 to 419: Each myofiber is innervated by one single NMJ, whose numbers should correspond to myofiber numbers in the normal muscle. But in this study, NMJ numbers were not always consistent with myofiber numbers at the same time point in the same muscle. For example, at the stage of NMJ increase, NMJ numbers were slightly fewer than myofiber numbers. This inconsistency might be the result of myofiber formation beginning earlier than NMJ formation. However, given that newly formed myotubes are innervated immediately by the axonal terminal to form NMJ^{48,49}, we speculated that the data bias generated by different quantitative methods for NMJs in 3D and myofibers in 2D sections possibly provides a reason for this inconsistency. The gastrocnemius in P9 and adult mice had more NMJ numbers (P9: 11676 ± 149 ; adult: 12089 ± 175) than myofiber numbers (P9: 10296 ± 314 ; adult: 10345 ± 297). Generally, during NMJ formation, many small and primitive AChR clusters form on myotubes (muscle pre-patterning)⁵⁰. During NMJ number quantification via AChR cluster counting, ectopic aneural AChR clusters may lead to the overestimation of NMJ numbers.

However, NMJ formation in the gastrocnemius of P9 and adult mice was over, and aneural AChR clusters had disappeared⁵¹, and this did not affect the accurate quantification of NMJ number. Thus, we speculated that the major reason for the inconsistency between NMJ and myofiber numbers in the gastrocnemius of P9 and adult mice is the underestimation of myofiber number due to the special anatomy of the gastrocnemius. Some myofibers in certain compartments of the gastrocnemius are pinnately distributed⁵², with myofiber numbers in the transverse section of the muscle belly fewer than the total number in the whole muscle.

2. *Also, the conceptual contribution of the experiments aimed at determining cell proliferation (line 148 onwards) seems not accurately presented or discussed. Although I agree with the idea that these experiments “should provide a quantifiable parameter for the determination of muscle development” it is not clear from the manuscript how this data could contribute to discriminating between muscle fiber hypertrophy and hyperplasia.*

Response: Thank you for your suggestions. According to your suggestions, we added some discussion.

Line 337 to 349: Skeletal muscle development depends primarily on the hypertrophy (size expansion) and hyperplasia (number increase) of myofibers. During myofiber hyperplasia, proliferative myoblasts exit the cell cycle after fusion³⁰. Therefore, the number change in proliferative cells was used to assess the developmental level of different muscles in this study. Alongside the myofiber number results, we found the existence of about 10%-20% of proliferative cells in three skeletal muscles when myofiber numbers developed to adult levels. Considering that myofiber hypertrophy requires an increase in myonuclei number within the first 21 days of birth¹¹, it can be speculated that these proliferative cells are necessary to provide new myonuclei for supporting myofiber hypertrophy. A percentage of proliferative cell numbers significantly higher than 20%, such as in the gastrocnemius on P0 and P3, points to the occurrence of not only myofiber hypertrophy but also myofiber hyperplasia. Thus, the quantitative analysis of proliferative cells not only compares the differences in developmental levels between different skeletal muscles but also provides a reference for determining muscle hypertrophy and hyperplasia.

3. *The article will benefit from the quantification of central nuclei in Fig. 2B.*

Response: Thank you for your suggestion. According to your suggestion, we analyzed the percentage of the myofibers with central nuclei in different muscles during postnatal development, and added quantitative result in **Figure 2C** and some description

Figure 2. Postnatal development of myofibers in different skeletal muscles.

Line 150 to 157: According to findings from the quantification of the percentage of myofibers with central nuclei (**Figure 2C**), central myonuclei rarely occupied the cleidomastoid areas after birth (4.60%), with most migrating to the periphery, pointing to the completion of myogenesis. As for the biceps brachii and gastrocnemius on P0, the percentages of myofibers with central myonuclei were 19.57% and 34.49%, respectively, indicating that some fibers in these two muscles were still immature myotubes. Subsequently, the central nuclei almost disappeared in the biceps brachii on P3 (3.93%) and in the gastrocnemius on P6 (2.38%), showing that myotubes in the biceps brachii mature before those in the gastrocnemius.

4. *As the methodology is one of the key contributions of this manuscript, some of the novel methods require a much more detailed description. See, for instance, the “Quantification of the intramuscular innervation pattern” and the “Measurement of NMJ numbers” descriptions.*

Response: Thank you for your suggestions. According to your suggestions, we added some more detailed description of quantification methods in **Materials and Methods**.

Line 655 to 660: The complete process is shown in **Figure S8**. First, the Filament module in the toolbar was selected, and the automatic creation was skipped for the manual editing mode (**Figure S8A**). Next, the Cone style was selected, and the AutoDepth method was used (**Figures S8B and S8C**). “AutoDepth” can recognize the signal in 3D space automatically when drawing a cone from the starting point to the endpoint. The signal depth is automatically calculated and recognized to locate the signal and trace continuous nerve fibers.

Figure S8. Quantification of the innervation pattern using Filament module in the Imaris.

Line 662 to 671: The entire process is shown in **Figure S9**. First, the Spots module was selected in the toolbar, and the calculation mode was scrutinized for the region of interest (**Figure S9A**). Then the size of the calculated region was set based on coordinates (**Figure S9B**), and the approximate diameter of the NMJ particle was entered (**Figure S9C**). Next, the algorithm “Intensity StdDev” was selected to identify the particle, and the number of identified particles was determined by adjusting the threshold range (**Figure S9D**); parameter values are obtainable based on image quality. Finally, the segmentation process was completed, and the “Add/Delete” mode was selected manually to correct the segmentation results (**Figure S9E**). With the use of artificial judgment, the unrecognized signal points were added, and incorrect data points from aneural AChR clusters, residual bubbles in samples, and the autofluorescence of muscle fibers and blood vessels were eliminated.

Figure S9. Quantification of NMJ number using Spots module in the Imaris.

5. *To help a better data conceptualization, please avoid purely methodological descriptions in the discussion (see, for instance, lines 339-342). Also, specific figures should not be cited in this section.*

Response: Thank you for your suggestion. According to the suggestion, we moved the methodological descriptions to the **Results (Line 238 to 241)**, and we also deleted the specific figures cited in the **Discussion**.

My next comments are as follows:

General:

6. *Although the manuscript is understandable, English correction is highly recommended, not only to correct many mistakes but mainly to improve the reading fluency of the manuscript.*

Response: Thank you for your suggestion. According to your suggestion, we have now worked on both language and readability and have also involved native English speakers for language corrections.

7. *Please reconsider the use of a black background in all figures as it does not seem to be always helpful.*

Response: Thank you for your suggestion. We have modified **Figure 4** by removing black background.

Figure 4. Intramuscular axonal arborization during postnatal development.

8. *When addressing the number of NMJs, it is important to mention that these quantifications are based mainly on postsynaptic AChR aggregates and to discuss the potential limitations of this choice.*

Response: Thank you for the suggestion. According to your suggestion, we added some discussions.

Line 396 to 400: During the quantification of the number of NMJs, the fluorescent signal of the postsynaptic AChR cluster was identified as a target for counting, probably resulting in the overestimation of NMJ numbers due to the potential existence of ectopic aneural AChR clusters during NMJ formation. Therefore, to ensure accurate counting, AChR cluster signals that did not overlap with the YFP signals of motor terminals, i.e., non-innervating AChR clusters, were excluded by manual correction.

Specific:

9. *There is no description for Fig. 2A regarding the muscle pictures.*

Response: Thank you for the suggestion. According to your suggestion, we added some descriptions for **Figure 2A**.

Line 139 to 142: We selected common experimental muscles in from different body parts, including the cleidomastoid, biceps brachii, gastrocnemius, and tibialis anterior for this study (**Figure 2A and Figure S1A**). The comparisons of the muscle sizes at different times after birth qualitatively demonstrated the rapid progression of early postnatal development.

10. *Please include statistical analyses of the data in Fig. 4B*

Response: Thank you for your suggestion. We added statistical analysis of the data in **Figure 4B**.

11. *Please be more specific in the description of the usefulness of addressing Collagen XXV levels in the context of this manuscript.*

Response: Thank you for your suggestion. Muscle-derived collagen XXV is indispensable for intramuscular innervation. And collagen XXV mRNA in skeletal muscle is only expressed during development, and gradually decreases with maturation. Thus, the quantification of collagen XXV mRNA expression in skeletal muscle can reflect the developmental level of intramuscular motor axonal terminals. The relevant description of the usefulness of quantifying collagen XXV mRNA expression has been written in **Results (Line 228 to 231)** and **Discussion (Line 364 to 371)**.

Line 228 to 231: Collagen XXV is transmembrane-type collagen that regulates axonal arborization and is indispensable for motor innervation in developing muscles³¹. Its mRNA is expressed in developing skeletal muscles and decreases gradually with the maturation of muscles, reflecting the progress of the axonal arborization of intramuscular motor nerves.

Line 364 to 371: Here, we focused on collagen XXV, transmembrane-type collagen, identified initially as a component of the senile plaque, amyloid of Alzheimer's disease of the brain⁴². Tanaka et al. demonstrated that without collagen XXV, motor axons successfully reach target muscles but fail to arborize to intramuscular branches⁴³. Moreover, muscle-derived collagen XXV mRNA is highly expressed in developing muscles, decreases with muscle development, and disappears almost entirely in adulthood^{31,43}.

12. *Please briefly explain the selection of the Shapiro-Wilk and Levene tests for the statistical analyses.*

Response: Thank you for your suggestion. According to your suggestion, we added some brief descriptions in **Statistical analysis**.

Line 671 to 673: "The Shapiro-Wilk test, a method for small sample sizes, was used to assess the normality of data distribution in each experiment. The heterogeneity of variance was evaluated using the Levene test, a stable method for both normally and non-normally distributed data....."

REVIEWERS' COMMENTS:

Reviewer #1 (Remarks to the Author):

The authors have done a great job at responding to all reviewer comments and thus I am now happy to accept.

Reviewer #2 (Remarks to the Author):

The authors have responded constructively to all of my concerns. I am thus happy for the article to be accepted.